# Can interplanetary magnetic field reach the Venus surface?

Yasuhito Narita[1,2] and Uwe Motschmann[3,4]

[1]Space Research Institute, Austrian Academy of Sciences, Schmiedlstr. 6, A-8042 Graz, Austria
[2]Institute of Physics, University of Graz, Universitätsplatz 5, A-8010 Graz, Austria
[3]Institut für Theoretische Physik, Technische Universität Braunschweig, Mendelssohnstr. 3, D-38106 Braunschweig, Germany
[4]Deutsches Zentrum für Luft- und Raumfahrt, Institut für Planetenforschung, Rutherfordstr. 2, D-12489 Berlin, Germany

**Correspondence:** Y. Narita
(yasuhito.narita@oeaw.ac.at)

**Abstract.** The question is addressed if there is a possibility of interplanetary magnetic field reaching the Venus surface by magnetic diffusion across the ionosphere. We present a model calculation and estimate the magnetic diffusion time at Venus, and find out that the typical diffusion time scale is in a range between 12 hours and 54 hours, depending on the solar activity and the ionospheric magnetic field condition. Magnetic field can thus permeate Venus surface and even Venus interior when
the solar wind is stationary (i.e., no magnetic field reversal) on the time scale of half-a-day to several days.

## 1   Problem of Venus surface magnetic field

Venus, being the nearest neighbor to the Earth, differs from the Earth in that the intrinsic magnetic field is absent. Nevertheless, a magnetospheric cavity is formed around Venus with a standing shock wave (bow shock) and a magnetotail as the solar wind becomes deflected by the Venus ionosphere and the interplanetary field drapes around the planet. In situ measurements by
Pioneer Venus Orbiter studied the Venus magnetic environment in detail such as a tail structure (Saunders and Russell, 1986) or bow shock (Russell et al., 1988).

Low-altitude profile of the Venus magnetic field was first obtained during the Pioneer Venus Orbiter entry in the nightside ionosphere (Russell et al., 1993). The magnetic field becomes stronger above an altitude of about 160 km. Overall, the magnetic field is in the range between 10 nT and 50 nT. Venus Express magnetometer (Zhang et al., 2006) further observed the Venus
magnetic field at altitudes of as low as 130 km over the Venus north pole during the aero-braking campaign. The average field is about 45 nT from an altitude of 300 km down to 180 km (Zhang et al., 2015) with a peak of about 90 nT at 200 km. Further down, the field magnitude decreases from 12 nT at an altitude of 150 km to 7 nT at an altitude of 130 km (Zhang et al., 2016).

We address the question if the magnetic field of interplanetary origin can ever reach the Venus surface. Hybrid code simulations, for example in Martinecz et al. (2009), suggest a penetration of the atmosphere by the interplanetary magnetic field
in less than an hour. Typical time scale for the magnetic field penetration is estimated from the hybrid plasma simulation by taking the total simulation time (not the computation time) as an upper limit. The total simulation time represents the time by which the magnetosphere (or induced magnetosphere) reaches a quasi-stationary state and the interplanetary magnetic field penetrates the ionosphere. The penetration time (using the total simulation **time** as proxy) is about 1000 s at Venus (Martinecz,

2008) and about 1400 to 1800 s at Mars (Bößwetter et al., 2004; Bößwetter, 2009). As the grid resolution is not very high, the **numerical resistivity significantly exceeds physical resistivity.** Thus the simulated penetration time may not be taken as very accurate, and improvements of the model are appropriate. Numerical diffusion cannot be avoided in the numerical simulation studies, and the diffusion time estimate may not be realistic in the simulation studies. Moreover, the hybrid plasma simulation code treats electrons as a massless fluid and the electron-neutral collisions are not included. Therefore, our theoretical calculation is complementary to the numerical studies on the diffusion problem. A more recent hybrid simulation study indicates that magnetic diffusion may be taking place in the ionosphere during **an ICME** (interplanetary coronal mass ejection) event at Venus (Dimmock et al., 2018).

To answer the question on the magnetic field at the Venus surface, we estimate the magnetic diffusion time in the Venus atmosphere. Two competing scenarios are possible. In scenario 1, naively speaking, the magnetic field can reach the planetary surface and even penetrate the planetary body, which is achieved when the Venus atmosphere is sufficiently diffusive and the interplanetary magnetic field surrounding Venus is stationary for a longer period of time. In scenario 2, on the other hand, the magnetic diffusion process at low altitudes becomes reset when the external field (in the induced magnetic field) reverses its orientation. Here, we mean by the "reset" a change in the sunward or anti-sunward direction of the interplanetary magnetic field. Since the diffusive transport process is local and **linear in the magnetic field,** the diffusive transport problem is not affected by the amount of **magnetic energy supplied to the ionosphere.**

The problem of the surface magnetic field at Venus is formulated as a competition between the diffusion time (such that the field reaches the surface on a detectable level) and the reset time (such that the field diffusion process is reset by the change in the interplanetary magnetic field). The interplanetary magnetic field has a four-sector structure in the solar ecliptic plane in the solar minimum phase. Therefore, the longest time length for a stable interplanetary magnetic field (without the field reversal due to the sector boundary crossing) is about 6 to 7 days. We take the four-sector structure of the interplanetary magnetic field (IMF) **to infer** the longest time interval (as the upper time limit) of the stable IMF. There is no large-scale pattern known to the Venus induced magnetosphere unlike the Earth substorm case. Solar minimum is more relevant to our theoretical model because the four-sector structure holds well and the coronal mass ejection (CME) occurrence rate **(which shortens the time length for the stable IMF) is minimum.**

Here we find that the magnetic diffusion time in the Venus atmosphere is of the order of **44,000 to 194,000 s, that is, in the range between 12 hours and 54 hours.** It is thus likely that the interplanetary magnetic field reaches the Venus surface and further into the Venus interior for a long time period of stationary solar wind. Our conclusion will be tested against the upcoming magnetic field measurements of the low-altitude region **(down to 1000 km) during the BepiColombo flyby at Venus.**

It is worth mentioning that the convective transport of the magnetic field is also an important transport mechanism, and the magnetic Reynolds number gives an estimate of the ratio of the convective transport to the diffusion. However, our study works on a more simplified situation to give an estimate by reducing the convective-diffusive **problem to** a diffusive problem. The reason for this is that the convective transport does not enter the problem of the vertical diffusion (in the sense of radial direction from the planet) and the plasma flow is in the horizontal direction (tangential to the planet surface). The convective

transport makes the penetration time longer, and not shorter. Therefore, our study gives an estimate of the lower limit (i.e., the shortest time) of the magnetic field penetration through the ionosphere.

## 2 Diffusion time estimate

### 2.1 Order of magnitude

We first estimate the diffusion time in an order-of-magnitude fashion. Magnetic diffusion time $\tau_{\mathrm{d}}$ is defined as

$$\tau_{\mathrm{d}} = L^2 \mu_0 \sigma, \tag{1}$$

where $L$ is the characteristic length scale, $\mu_0 = 4\pi \times 10^{-7}$ H m$^{-1}$ is the permeability of free space, and $\sigma$ is the electric conductivity. The Pedersen conductivity is relevant to the diffusion problem here.

**In general, conductivity in a magnetized plasma is a tensor, whose components are** (1) Pedersen conductivity, (2) Hall
conductivity, and (3) field-aligned or parallel conductivity. **Above all, the Pedersen conductivity is relevant to the problem of diffusion time estimate. The reason for this is that magnetic diffusion takes time because energy is dissipated along the way (magnetic energy is converted to heat). It is the Pedersen current by which the energy dissipation is achieved. The Hall current, in contrast, has no energetic effect. From a geometrical point of view,** the Pedersen conductivity (or the current, to be more precise) can transmit the magnetic field (say, in the $x$-direction in the horizontal plane **or in the current-**
**carrying layer of the ionosphere)** by the electric current flowing perpendicular to the magnetic field (in the $y$-direction in the horizontal plane) and generate the magnetic field in the same direction to the original magnetic field (in the $x$-direction) by Ampère's law on the opposite side of the current layer (on the ground or low-altitude side of the current layer). The Hall current cannot transfer the magnetic field across the current layer because the current direction is pointing vertically. The parallel current cannot transfer the field in a homogeneous fashion, either. The parallel current (in the $x$-direction) can generate
the magnetic field across the current layer but the field rotates into the minus $y$-direction below the current layer. It is also worth while to note that the Pedersen conductivity also converts the magnetic energy into heat.

We take the length scale (or thickness in altitude) $L = 100$ km for the conducting atmospheric layer and the Pedersen conductivity of about $\sigma =1$ S m$^{-1}$ (justified in section 2.2). We obtain the diffusion time of the order of 10,000 s (more exactly, 12,566 s when using the nominal values above). The magnetic field can thus penetrate the Venus ionosphere within about
200 minutes (or about 3.5 hours). As we see below, the conductivity can be even higher by one order of magnitude, and the diffusion time scales up to 100,000 s.

### 2.2 More quantitative estimate

In reality, the conductivity depends on the altitude, the ionospheric condition, and the solar activity. We estimate the diffusion time more quantitatively by numerically integrating the differential diffusion time $L\mu_0\sigma$ over the altitudes in the following

way:

$$\tau_{\mathrm{d}} = \int_{z_{\min}}^{z_{\max}} 2z\mu_0\sigma\,\mathrm{d}z \tag{2}$$

$$= L^2\mu_0\langle\sigma\rangle, \tag{3}$$

where $z$ is the altitude from the surface, $z_{\min}$ and $z_{\max}$ the lower and upper limits of the height integration, $L = z_{\max} - z_{\min}$
the thickness of the diffusion layer, and $\langle\sigma\rangle$ the average conductivity The factor 2 in the integration comes from the fact that
the integration yields $L^2\mu_0\sigma$, if the conductivity is constant over the altitude change.

The task is to evaluate the electric conductivity as a function of the altitude. Since we work on the Pedersen conductivity
for the magnetic diffusion problem, the electron density, the collision frequency, and the magnetic field profiles are needed to
calculate the conductivity before performing the height integration. The procedure of the diffusion time estimate is summarized
as follows.

1. Electron density profile.

   Altitude-dependent electron number density data are obtained by the Pioneer Venus Orbiter radio occultation measurements. We take values from Figure 2 in Kilore and Luhmann (1991) at higher solar zenith angles (above 55 degree),
   under the condition of solar maximum and that of solar minimum. The profile of the electron density is displayed in the
first panel of Fig. 1.

2. Collision frequency profile.

   The profile of the collision frequency is taken from the recent calculation by Dubinin et al. (2014) (Figure 16 in the
   article) which is based on theoretical velocity-moment estimates (Schunk and Nagy, 2000) using the temperature and
   neutral density profiles from Fox and Sung (2001). We consider the electron-neutral collisions **and the ion-neutral**
**collisions in the present work.** The collision frequency is displayed as a function of the altitude in the second panel of
   Fig. 1.

3. Magnetic field profile.

   Magnetic field data from Venus Express are used as a reference from 300 km to 180 km (Villarreal et al., 2015) and
   further down to 130 km (Zhang et al., 2016). The former data set is from a single event, but is illustrative in the model
construction in that the transition is smooth with a magnetic pileup and an asymptotic behavior at higher altitudes (solid
   curve in black above an altitude of 170 km in the third panel of Fig. 1). The latter data set is from 33 peri-apsis passages,
   and we take the median values (solid curve in black below 150 km in the same panel). We use the secant function
   $\mathrm{sech}(x) = 2/(\exp(x) + \exp(-x))$ to construct a magnetic field model. The secant function is used separately below the
   magnetic pileup peak (set to $z_0 = 200$ km altitude) and above the peak in the form of $B = B_1\mathrm{sech}((z - z_0)/d) + B_0$.
We use the secant function as an empirical model because the secant function is known to describe solitary structures
   such as the KdV soliton (Korteweg-de Vries) or the density profile of the Harris current sheet. We obtain from the fitting

procedure the following coefficients: (1) $B_0 = 40.0$ nT (offset value), $B_1 = 50.0$ nT (height of the secant bell shape), and $d = 7.0$ km (width of the bell shape) for $z_0 \geq 200$ km, and (2) $B_0 = 6.5$ nT, $B_1 = 83.5$ nT, and $d = 12.0$km for $z_0 \leq 200$ km. The uncertainty of the magnetic field model is inferred from the statistical fluctuations shown in Zhang et al. (2016), and is approximated to a factor of $0.5$ for the lower limit and a factor of $1.5$ for the upper limit. Graphics of the magnetic field model are displayed in the third panel of Fig. 1.

4. Pedersen conductivity.

We take the Pedersen conductivity (Vasyliunas, 2012; Dubinin et al., 2014):

$$\sigma_{\mathrm{p}} = n_{\mathrm{e}}e^2 \left( \frac{\nu_{\mathrm{in}}}{m_{\mathrm{i}}(\nu_{\mathrm{in}}^2 + f_{\mathrm{gi}}^2)} + \frac{\nu_{\mathrm{en}}}{m_{\mathrm{e}}(\nu_{\mathrm{en}}^2 + f_{\mathrm{ge}}^2)} \right), \tag{4}$$

where $n_{\mathrm{e}}$ is the electron number density, $e$ the elementary charge, $m_{\mathrm{i}}$ and $m_{\mathrm{e}}$ the mass of ions (assuming protons) and electrons, respectively, $f_{\mathrm{gi}}$ and $f_{\mathrm{ge}}$ the gyrofrequency of ions and electrons (in units of s$^{-1}$, not the angular frequency in units of rad s$^{-1}$), $\nu_{\mathrm{in}}$ the collision frequency between ions and neutrals, and $\nu_{\mathrm{en}}$ the collision frequency between electrons and neutrals.

The ion term in the Pedersen conductivity (Eq. 4) should not be neglected because the ratio of the collision frequency to the respective (electron or ion) gyro-frequency is not neglibible for the electrons and the ions. For example, Dubinin et al. (2014) show that the ion-neutral collision frequency exceeds the ion gyro-frequency at altitudes below 220 km. In contrast, the electron-neutral collision frequency exceeds the electron gyrofrequency at altitudes below 140 km. We evaluate the conductivity by keeping the ion term in Eq. (4) in the calculation. The dominant ion species is not protons but heavier species such as oxygen atoms O$^+$ or molecules O$_2^+$ (Fox and Sung, 2001). We choose for the follwoing mean ion masses: 11.6 proton mass or 23.3 proton mass, values taken from Dubinin et al. (2014). The Pedersen conductivity is displayed as a function of the altitude in the fourth panel of Fig. 1 for the solar maximum and minimum, respectively, including both the magnetic field models (high-field, mean-field and low-field) and the ion mass models. The peak conductivity is in the range at about 10 S m$^{-1}$.

5. Integration.

Height integration is performed to evaluate the diffusion time $\tau_{\mathrm{d}}$ using Eq. (2) and the five-point Newton-Cotes integration formula. We resample values of the electron density, the collision frequency, and the model magnetic field for the numerical integration at a spatial resolution of 1 km, and extrapolate the values in a linear fashion on the logarithmic scale at altitudes down to 100 km and up to 400 km. **Most of the conductance (height-integrated conductivity) is confined around the maximum at 150 km. The contribution of this current-carrying layer from 140 km to 170 km in altitude is around 78–85 % during the solar maximum (the variation comes from the choice of the magnetic field mode and the ion profile) and 88–91 % during the solar minimum.**

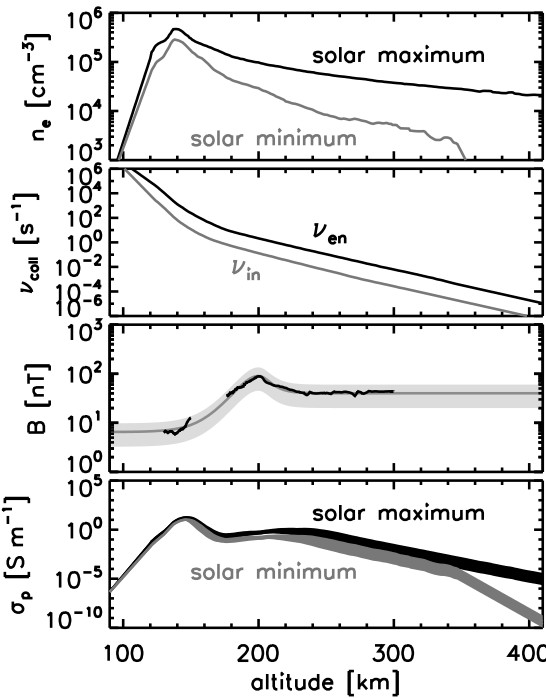

**Figure 1.** Electron number density $n_e$ from Pioneer Venus Orbiter radio occultation measurement after Kilore and Luhmann (1991), model collision frequency between electrons and neutral particles $\nu_{en}$ **and that between ions and neutral particles** $\nu_{in}$ after Dubinin et al. (2014), magnetic field $B$ after Venus Express measurements in black after Villarreal et al. (2015); Zhang et al. (2016) and model magnetic field with a fluctuation range (in gray), and Pedersen conductivity $\sigma_p$ as a function of altitude from the Venus surface.

**Table 1.** Results from the diffusion time estimate. The symbol $\tau_d$ stands for the diffusion time. The range in the table represents the choice for the mean ion mass (11.6 proton mass or 23.3 proton mass, values taken from Dubinin et al. (2014).

| magnetic field model | $\tau_d$ at solar maximum | $\tau_d$ at solar minimum |
|---|---|---|
| mean field case | 109,068 – 114,728 s | 58,811 – 59,202 s |
| strong field case | 84,325 – 85,441 s | 44,698 – 44,906 s |
| weak field case | 194,268 – 194,343 s | 92,288 – 93,536 s |

## 2.3 Results of diffusion time estimate

Diffusion time varies in the range from about 44,000 s (about 12 hours) to about 194,000 s (54 hours). Solar activity and the local magnetic field in the ionosphere influence the diffusion time. A minimum of 12 hours (half-a-day) for the diffusion time is needed for the magnetic field to penetrate the Venus ionosphere and atmosphere. If the electron density is higher or the local magnetic field weaker, the diffusion time can scale up to 54 hours (more than 2 days). Therefore, Venus surface may exhibit

non-zero magnetic field when the solar wind is stationary (in the sense that the interplanetary magnetic field does not show a reversal) on the time scale of half-a-day to several days. **For reference, we repeat the calcuation of the diffution time using only the electron term in the conductivity. The diffusion time from the electron term is in the range from about 40,000 s (11 hours) to about 146,000 s (40 hours). The ion contribution makes a difference in the diffusion time by about 10– 20 %. Note that the peak Pedersen conductivity is about $14\,\mathrm{S\,m^{-1}}$ from the electron term and about 0.035 S m$^{-1}$ (Tab. 2 right column), i.e., the ion contribution is only about 0.1 % at the peak of the Pedersen conductivity height-profile.**

## 3  Discussion

### 3.1  Comparison with the hybrid simulations

**It is interesting to observe the difference between the simulation and analytic estimates of the diffusion time by about two orders of magnitude. The major reason behind this difference most likely lies in the neglect of the electron-neutral collisions and the numerical diffusion in the hybrid simulations.**

– **Electron contribution. The ion term in the conductance estimate contributes to a longer diffusion time by 10– 20 %. In other words, the electron contribution to the conductance (and diffusion time) is somewhat 80–90 %, i.e., by neglecting this contribution the diffusion time is 5 to 10 times shorter than its actual value. Therefore, considering the electron contribution would increase the simulation proxy of 1000 s to 5000–10000 s, i.e., 1.5–3 hours. This reduces the difference between the simulation and analytic result by one order of magnitude.**

– **Numerical diffusion. Problem of numerical oscillation, rounding or cutoff error, and numerical instabilities in the computation are suppressed either by adding an articicial diffusion or imposing a spatial smoothing (Winske and Omidi, 1993; Bagdonat, 2002). Here we estimate the diffusion coefficient or the conductivity for the smoothing procedure. First, we rewrite the magnetic diffusion equation**

$$\partial_t \boldsymbol{B} = \eta \nabla^2 \boldsymbol{B} \tag{5}$$

**into the time advancing formula as**

$$
\begin{aligned}
\boldsymbol{B}(t+\Delta t) &= \eta \nabla^2 \boldsymbol{B}(t)\,\Delta t \tag{6}\\
&\simeq \frac{\eta}{\ell^2}\boldsymbol{B}(t)\,\Delta t, \tag{7}
\end{aligned}
$$

**where $\eta$ denotes the diffusion coefficient of the magnetic field, $B$ is the magnetic field, $\Delta t$ is the time step in the simulation, and $\ell$ is the length scale of the gradient. Second, in the smoothig method, the magnetic field is smoothed by the following procedure (Winske and Omidi, 1993; Bagdonat, 2002; Mülle et al., 2011),**

$$\boldsymbol{B}(t) \to \boldsymbol{B}(t) - \alpha_{\mathrm{sm}}\left(\boldsymbol{B}(t) - \langle \boldsymbol{B} \rangle\right), \tag{8}$$

where $\alpha_{\rm sm}$ is a free parameter called the the smoothing factor (its value of $\alpha_{\rm s}$ is typically 0.01 to 0.1) and $\langle B \rangle$ is the locally-smoothed magnetic field. Now, by comparing the right-hand side of Eq. (7) with the smoothing term $\alpha_{\rm sm} B$ in Eq. (8), we obtain a relation between the smoothing factor $\alpha_{\rm sm}$ and the diffusion coefficient $\eta$ as follows,

$$\eta = \frac{\alpha_{\rm sm} \ell^2}{\Delta t}. \tag{9}$$

Using the relation to the conductivity $\eta = (\mu_0 \sigma)^{-1}$, Eq. (9) is expressed as

$$\sigma = \frac{\Delta t}{\mu_0 \alpha_{\rm sm} \ell^2}. \tag{10}$$

We evaluate Eq. (10) using the following values: the smoothing parameter $\alpha_{\rm sm} = 0.01$ (i.e., 1 % spatial smoothing) from Mülle et al. (2011), $\ell = 100$ km (grid size in the simulation) from Bößwetter et al. (2004) and Martinecz et al. (2009), $\Delta t = 1$ s (typical ion gyroperiod in the solar wind), and $\mu_0 = 4\pi \times 10^{-7}$ H m$^{-1}$ (permeability of free space), and obtain the numerical conductivity (or the smoothing conductivity) $\sigma_{\rm sm} = 7.96 \times 10^{-3}$ S m$^{-1}$. Hence, the numerical conductivity equivalent to the spatial smoothing procedure for the purpose of numerical damping in the hybrid simulation is of the order of $10^{-2}$ S m$^{-1}$. The physical conducticity from our study is 1–10 S m$^{-1}$ at the peak.

The magnetic diffusion time is $L^2 \mu_0 \sigma$ and the diffusion length scale $L$ is the same as the grid scale $l$. Hence, the diffusion time is proportional to the conductivity in our problem, and the difference in the diffusion time by two orders of magnitude between the hybrid simulations (about 1000 s) and our semi-analytic estimate (about 100,000 s) can reasonably be explained by the spatial smoothing procedure in the simulation.

### 3.2 Comparison with the Earth ionosphere

It is also interesting to observe that the Pedersen conductivity at the Venus is much higher than at the Earth by about four orders of magnitude. This difference can be explained as follows. We write formulas separately for the electron Pedersen conductivity $\sigma_{\rm p,e}$ S m$^{-1}$ as

$$\sigma_{\rm p,e}[{\rm Sm}^{-1}] \quad = \quad \frac{n_e e^2}{m_e} \frac{\nu_{\rm en}}{\nu_{\rm en}^2 + f_{\rm ge}^2} \tag{11}$$

$$\simeq \quad 2.8 \times 10^{-2} \times n_e[{\rm cm}^{-3}] \times \frac{\nu_{\rm en}[{\rm Hz}]}{(\nu_{\rm en}[{\rm Hz}]) + (f_{\rm ge}[{\rm Hz}])^2} \tag{12}$$

and the ion Pedersen conductivity $\sigma_{\rm p,i}$ S m$^{-1}$ as

$$\sigma_{\rm p,i}[{\rm Sm}^{-1}] \quad = \quad \frac{n_e e^2}{m_i} \frac{\nu_{\rm in}}{\nu_{\rm in}^2 + f_{\rm gi}^2} \tag{13}$$

$$\simeq \quad 1.3 \times 10^{-6} \times n_e[{\rm cm}^{-3}] \times \frac{\nu_{\rm in}[{\rm Hz}]}{(\nu_{\rm in}[{\rm Hz}])^2 + (f_{\rm gi}[{\rm Hz}])^2}, \tag{14}$$

where a mass ratio of $m_i/m_p = 11.6$ or $m_i/m_e = 2.1 \times 10^4$ is used ($m_i$ is the ion mass, $m_p$ is the proton mass, and $m_e$ is the electron mass).

The Pedersen conductivity at the Venus $\sigma_{\mathrm{p}}^{(\mathrm{V})}$ is primarily contributed by the electron-neutral colissions. By ignoring the gyro-frequency, the Venus Pedersen conductivity is approximately as follows.

$$\sigma_{\mathrm{p}}^{(\mathrm{V})} \sim \frac{n_{\mathrm{e}} e^2}{m_{\mathrm{e}} \nu_{\mathrm{en}}} \tag{15}$$

The Pedersen conductivity at the Earth $\sigma_{\mathrm{p}}^{(\mathrm{E})}$ is, in contrast to the Venus case, contributed by the ion-neutral collisions.

$$\sigma_{\mathrm{p}}^{(\mathrm{E})} \sim \frac{n_{\mathrm{e}} e^2}{m_{\mathrm{i}} \nu_{\mathrm{in}}} \tag{16}$$

The reason for this is that the gyro-frequency exceeds the electron-neutral collision frequency at an altitude of 60–70 km and above due to a stronger magnetic field (than that of the Venus). Now we compare Eq. (15) with Eq. (16).

Using the facts that (1) the electron density is almost the same ($n_{\mathrm{e}} \sim 10^5 \, \mathrm{cm}^{-3}$) between the Venus (peak altitude $z = 150 \, \mathrm{km}$) and the Earth (peak altitude $z = 130 \, \mathrm{km}$), (2) the typical mass ratio from the ions to the electrons is about 20,000 (about 11 proton mass), and (3) the collision frequency is rougfhly of the same order beween the Venus and the Earth, $\nu_{\mathrm{en}}^{(\mathrm{V})} = 250 \, \mathrm{Hz}$ and $\nu_{\mathrm{in}}^{\mathrm{E}} = 50 \, \mathrm{Hz}$ at the Earth, respectively, we obtain the ratio of the peak Pedersen conductivity from the Venus to the Earth as follows:

$$\frac{\sigma_{\mathrm{p,e}}^{(\mathrm{V})}}{\sigma_{\mathrm{p,i}}^{(\mathrm{E})}} \sim \frac{m_{\mathrm{i}}}{m_{\mathrm{e}}} \frac{\nu_{\mathrm{in}}^{(\mathrm{E})}}{\nu_{\mathrm{en}}^{(\mathrm{V})}} \sim 10^4. \tag{17}$$

The difference in the peak Pedersen conductivity by nearly four orders of magnitude can essentially represent the difference in the Pedesen current carrier in the different magnetic field environments: the Pedersen current is carried by the electrons at the Venus and by the ions at the Earth. More detailed calculations of the peak Pedersen conductivity at the Earth and the Venus are shown in Tab. 2.

## 4   Concluding remark

We conclude the diffusion time estimate with the following notes. First, a stationary solar wind condition on a time scale of half-a-day to several days is likely occurring in the Venus environment. The interplanetary magnetic field can theoretically reach (under the condition of stationary solar wind) the Venus surface and justifies the non-zero field measurements by Venus Express. Second, further improvement is possible by including the ion-neutral collisions and the solar activity influence on the collision frequency. Third, the upcoming missions such as Parker Solar Probe (Fox et al., 2016), BepiColombo (Benkhoff et al., 2010), and Solar Orbiter (Müller et al., 2013) will perform magnetic field and plasma measurements in the near-Venus environment **for a variety** of distances and approaching directions to Venus. For **example, two Venus flyby manoeuvres are planned for BepiColombo:** Flyby 1 in October 2020 **down to** 11,317 km, and Flyby 2 in August 2021 down to 1,000 km. Even though BepiColombo's flybys at Venus are too far to directly measure the near-surface magnetic field, the flyby data will help us to determine or constrain the stability of IMF and the condition for the magnetic field penetration through the ionosphere **for several hours to a day.** The direct test for the magnetic field penetration would ideally be performed during a stable IMF period, for another Venus mission in future.

**Table 2. Comparison of Pedersen conductivity estimate at the peak altitudes of the conductivity, $z = 130$ km at the Earth and $z = 150$ km at the Venus. Electron density value at the Earth is taken from Kelly (1989) at $z = 130$ km. Electron density value at the Venus is from Kilore and Luhmann (1991) during the solar maximum at an altitude of $z = 150$ km (at the peak of Pedersen conductivity). Electron gyro-frequency is calculated from the nominal magnetic field magnitude. Collision frequency values at the Earth are from Kertz (1989) and that at the Venus are from Dubinin et al. (2014). A mean mass ratio of $11.6$ is used between the ions and the protons.**

|  | Earth | Venus |
|---|---|---|
| magnetic field $B$ | $3.0 \times 10^5$ nT | $1.0 \times 10$ nT |
| electron density $n_\mathrm{e}$ | $1.2 \times 10^5$ cm$^{-3}$ | $2.9 \times 10^5$ cm$^{-3}$ |
| electron gyro-frequency $f_\mathrm{ge}$ | $8.4 \times 10^4$ Hz | $2.8 \times 10^2$ Hz |
| electron-neutral collision frequency $\nu_\mathrm{en}$ | $1.3 \times 10^3$ Hz | $2.5 \times 10^2$ Hz |
| electron Pedersen conductivity $\sigma_\mathrm{p,e}$ | $6.2 \times 10^{-4}$ S m$^{-1}$ | $1.4 \times 10^1$ S m$^{-1}$ |
| ion gyro-frequency $f_\mathrm{gi}$ | $3.9 \times 10^1$ Hz | $1.3 \times 10^{-2}$ Hz |
| ion-neutral collision frequency $\nu_\mathrm{in}$ | $5.0 \times 10^1$ Hz | $1.1 \times 10$ Hz |
| ion Pedersen conductivity $\sigma_\mathrm{p,i}$ | $2.3 \times 10^{-3}$ S m$^{-1}$ | $3.5 \times 10^{-2}$ S m$^{-1}$ |

*Competing interests.* The authors declare that there is no competing interests.

*Acknowledgements.* The authors thank stimulating and fruitful discussion at the Europlanet workshop *Planetary Atmospheric Erosion* in Murighiol, Romania, 11–15 June 2018, hosted by Institute for Space Sciences in Bucharest-Măgurele. YN thanks Daniel Heyner for the information about the BepiColombo **Venus** flyby plan.

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
