# Peer review of "Can interplanetary magnetic field reach the Venus surface?"

_Annales Geophysicae, 2018_

## Referee Comment (RC1) · Anonymous Referee #1 · 13 Aug 2018

The paper *Can interplanetary magnetic field reach the Venus surface?* provides an estimate of the interplanetary magnetic field (IMF) diffusion time across the atmosphere of Venus – to be compared with previous simulation work, apparently underestimating it, and with upcoming observations by BepiColombo. The estimation is based on observed profiles of relevant quantities and numerical integration. I see the paper as a timely contribution, however, a few questions should be explained in more detail before publication.

1. Convective transport of the magnetic field is completely ignored. Is this related to the impact of the IMF draping around the planet on the magnetic Reynolds number? Please comment.

2. I do not fully understand the 'reset' of the magnetic field (p. 2, L5-6). Once the magnetic field enters the diffusive environment, i.e. the conductive ionosphere, the diffusive transport is local: If magnetic energy is supplied in sufficient amount to overcome ohmic losses, the IMF will make it to Venus surface. On the other hand, perhaps at that time the IMF has changed and the diffused magnetic field is no longer equal to the original IMF. Please clarify.

3. Along the same line, the 'reset' is discussed only in terms of the four-sector structure of the IMF. What about variability on top of this large scale pattern (which is essential e.g. for magnetic activity at Earth)? Does it matter less for Venus, in particular at solar minimum?

4. Please explain briefly why 'Pedersen conductivity transmits the magnetic field by diffusion' (p.3, L4-5). Also, phrasing could be perhaps adjusted a bit, since Pedersen current actually converts the magnetic energy into heat (see also 2).

5. The Pedersen conductivity is attributed to electrons (Eq. 5), partly because electron-neutral collision frequency is much larger than ion-neutral collision frequency. However, the conductance formula can be cast as ne/B ($\nu_{en}/f_{ge}/(1+\nu_{en}^2/f_{ge}^2)$ + similar ion term), therefore what actually matters is the ratio of the collision frequency to the respective (electron or ion) gyro-frequency. At Earth, most of the Pedersen current is carried in the E ionospheric layer by ions, while electrons, already non-collisional at E heights, carry the Hall current. Please comment the case at Venus in more detail.

6. The difference between the estimated diffusion time and the simulation result is larger than an order of magnitude. Please discuss this more closely. Incidentally, I was no able to find the less than one hour estimate (p. 1, L19) in the study of Martinecz et al. (2009).

7. The suggested test by BepiColombo relies on the stability of the IMF during the flyby (see also 2 and 3 above). Please comment.

8. Minors:
 - p. 1,  L9: about => around ?
 -        L12: Orbiter
 -        L14: studied => observed ?
 -        L16-17: The field magnitude becomes diminished => Further down, the field magnitude decreases
 -        L21: as a proof => as very accurate ?
 - p. 2, L3: becomes reset when the external field (in the induced magnetic field) reverses its orientation => please rephrase (see also 2).
 -        L8: report our study => find ?
 -        L10: delete 'by permeability'. For the substance of the message, see also 2, 3.
 -        L17: L is,…, $\mu_0$ … is, and $\sigma$ is…
 -        L20: strictly => exactly
 -        L25: I think '(or diffusion speed)' should be deleted.
 - p. 3,  L3: $z^2$ => $\Delta z^2$ (i.e. $z_{max}^2 - z_{min}^2$); $<\sigma>$ => $\sigma$ (no brackets); diffusivity => conductivity.
 -        L4-5: Please rephrase 'Since the Pedersen conductivity transmits the magnetic field by diffusion' (see also 4).
 -        L20: from Venus Express
 -        L31: referred => inferred

---

## Author Response (AR1)

Manuscript prepared for Ann. Geophys.
with version 4.2 of the LaTeX class copernicus.cls.
Date: 4 October 2018

**Reply to referee comments – angeo-2018-69**

Y. Narita[1,2] and U. Motschmann[3,4]

[1]Space Research Institute, Austrian Academy of Sciences, Schmiedlstr. 6, A-8042 Graz, Austria
[2]Institute of Physics, University of Graz, Universitätsplatz 5, A-8010 Graz, Austria
[3]Institut für Theoretische Physik, Technische Universität Braunschweig, Mendelssohnstr. 3,
D-38106 Braunschweig, Germany
[4]Deutsches Zentrum für Luft- und Raumfahrt, Institut für Planetenforschung, Rutherfordstr. 2,
D-12489 Berlin, Germany

*Correspondence to:* Y. Narita
(yasuhito.narita@oeaw.ac.at)

- *The paper Can interplanetary magnetic field reach the Venus surface? provides an estimate of the interplanetary magnetic field (IMF) diffusion time across the atmosphere of Venus to be compared with previous simulation work, apparently underestimating it, and with upcoming observations by BepiColombo. The estimation is based on observed profiles of relevant*

5      *quantities and numerical integration. I see the paper as a timely contribution, however, a few questions should be explained in more detail before publication.*

     – Reply: Thank you.

- *1. Convective transport of the magnetic field is completely ignored. Is this related to the impact of the IMF draping around the planet on the magnetic Reynolds number? Please comment.*

10      – Reply: Yes, the convective transport of the magnetic field is an important effect in general, and the magnetic Reynolds number indeed gives an estimate of the ratio of the convective transport to the diffusion. However, our study works on a more simplified situation to give an estimate by reducing the convective-diffusive problem into a diffusive problem. The reason for this is that the convective transport does not enter the

15      problem of the vertical diffusion (in the sense of radial direction from the planet) and the plasma flow is in the horizontal direction (tangential to the planet surface). The convective transport makes the penetration time longer, and not shorter. Therefore, our study gives an estimate of the lower limit (i.e., the shortest time) of the magnetic field penetration through the ionosphere.

20      – We will add the above statement in section 1, after p. 2, L3.

*– 2. I do not fully understand the reset of the magnetic field (p. 2, L5–6). Once the magnetic field enters the diffusive environment, i.e. the conductive ionosphere, the diffusive transport is local: If magnetic energy is supplied in sufficient amount to overcome ohmic losses, the IMF will make it to Venus surface. On the other hand, perhaps at that time the IMF has changed and the diffused magnetic field is no longer equal to the original IMF. Please clarify.*

– Reply:

We mean by the "reset" a change in the sunward or anti-sunward direction of the interplanetary magnetic field. It is true that the diffusive transport is local and linear. That is, the energy budget problem does not apply here, since the magnetic diffusion process (or the equation) is linear to the magnetic field and the diffusion time is determined by the ionospheric condition only (by the length scale and the conductivity).

– We will add a clarification above to p. 2, L5–6.

*– 3. Along the same line, the reset is discussed only in terms of the four-sector structure of the IMF. What about variability on top of this large scale pattern (which is essential e.g. for magnetic activity at Earth)? Does it matter less for Venus, in particular at solar minimum?*

– Reply:

We mean by the "reset" a change in the sunward or anti-sunward direction of the interplanetary magnetic field and we take the four-sector structure of the IMF for the reason of the longest time interval (as the upper time limit) of the stable IMF. There is no large-scale pattern known to the Venus induced magnetosphere, unlike the Earth substorm case. Solar minimum is more relevant to our theoretical model because the four-sector structure holds well and the CME occurrence rate (which even shortens the time length for the stable IMF) is minimum during the solar minimum.

– We will add a clarification above to p. 2, L5–6, and around.

*– 4. Please explain briefly why Pedersen conductivity transmits the magnetic field by diffusion (p.3, L4–5). Also, phrasing could be perhaps adjusted a bit, since Pedersen current actually converts the magnetic energy into heat (see also 2).*

– Reply:

Agreed. We will phrase like this:

"There are three different kinds of conductivity in the plasma: (1) Pedersen conductivity, (2) Hall conductivity, and (3) field-aligned or parallel conductivity. The Pedersen conductivity (or the current, to be more precise) can transmit the magnetic field (say, in the x-direction in the horizontal plane) by the electric current flowing perpendicular to the magnetic field (in the y-direction in the horizontal plane) and generate the magnetic field in the same direction to the original magnetic field (in the x-direction) by Ampère's law

**Table 1.** Revised version of table 1.

| magnetic field model | $\tau_{\rm d}$ at solar maximum | $\tau_{\rm d}$ at solar minimum |
|---|---|---|
| mean field case | 109,068 – 114,728 s | 58,811 – 59,202 s |
| strong field case | 84,325 – 85,441 s | 44,698 – 44,906 s |
| weak field case | 194,268 – 194,343 s | 92,288 – 93,536 s |

on the opposite side of the current layer (on the ground or low-altitude side of the current layer). The Hall current cannot transfer the magnetic field across the current layer because the current direction is pointing vertically. The parallel current cannot transfer the field in a homogeneous fashion, either. The parallel current (in the x-direction) can generate the magnetic field across the current layer but the field rotates into the minus y-direction below the current layer. It is also worth while to note that the Pedersen conductivity also converts the magnetic energy into heat."

– We will add the explanation on p.3, L4–5 and around.

– *5. The Pedersen conductivity is attributed to electrons (Eq. 5), partly because electron-neutral collision frequency is much larger than ion-neutral collision frequency. However, the conductance formula can be cast as ne/B (\nu_en/f_ge/(1+\nu_en^2/f_ge^2) + similar ion term), therefore what actually matters is the ratio of the collision frequency to the respective (electron or ion) gyro-frequency. At Earth, most of the Pedersen current is carried in the E ionospheric layer by ions, while electrons, already non-collisional at E heights, carry the Hall current. Please comment the case at Venus in more detail.*

– Reply:

Yes! Thank you very much for pointing out that the ion contribution may not be negligible. Indeed, Dubinin et al. (JGR 2014) show that the ion-neutral collision frequency exceeds the ion gyrofrequency at altitudes below 220 km. In contrast, the electron-neutral collision frequency exceeds the electron gyrofrequency at altitudes below 140 km. We improved our estimate by keeping the ion term in Eq. (4) in the calculation, and we will not use the electron-term approximation (Eq. 5) any more. The conductivity (now including the ion term) is higher than our previous result. The diffusion time becomes consequently longer than our previous estimate by about 10 to 20%. The revised figure and table are attached to this reply. The range in the table represents the choice for the mean ion mass (11.6 proton mass or 23.3 proton mass, values taken from Dubinin et al. 2014).

[Figure]

**Fig. 1.** Revised version of figure 1.

– We will update the item 4 (Pedersen conductivity) on page 4 and replace Fig. 1 and Tab. 1 by our new results.

85   – *6. The difference between the estimated diffusion time and the simulation result is larger than an order of magnitude. Please discuss this more closely. Incidentally, I was no able to find the less than one hour estimate (p. 1, L19) in the study of Martinecz et al. (2009).*

– Reply:

Thank you for double-checking the paper by Martinecz et al. (2009).

90    Typical time scale for the magnetic field penetration is estimated from the hybrid plasma simulation by taking the total simulation time (not the computation time) as an upper limit. The total simulation time represents the time by which the magnetosphere (or induced magnetosphere) reaches a quasi-stationary state and the interplanetary magnetic field penetrates the ionosphere. The penetration time (using the total simulation as proxy)

95    is about 1000 s at Venus (Martinecz, 2008) and about 1400 to 1800 s at Mars (Bößwetter et al., 2004, Bößwetter, 2009).

The numerical diffusion cannot be avoided in the numerical simulation studies, and the diffusion time estimate may not be realistic in the simulation studies. Moreover, the hybrid plasma simulation code treats electrons as a massless fluid and the electron-neutral

100         collisions are not included. Therefore, we find our theoretical calculation complementary to the numerical studies on the diffusion problem.

– We will add the above statement in section 1 (p. 1, L19–20) and also the following literatures:

(1) Bößwetter, A., Bagdonat, T., Motschmann, U., and Sauer, K.: Plasma boundaries at
105         Mars: a 3-D simulation study, Ann. Geophys., 22, 4363–4379, doi:10.5194/angeo-22-4363-2004, 2004.

(2) Bößwetter, A.: Wechselwirkung des Mars mit dem Sonnenwind: Hybrid-Simulationen mit besonderem Bezug zur Wasserbilanz, p. 59, PhD thesis, Tech. Univ. Braunschweig, Braunschweig, Germany, https://publikationsserver.tu-braunschweig.de/receive/dbbs_mods_00028707,
110         2009.

(3) Martinecz, C.: The Venus plasma environment: a comparison of Venus Express ASPERA-4 measurements with 3D hybrid simulations, p. 71, PhD thesis, Tech. Univ. Braunschweig, Braunschweig, Germany, https://publikationsserver.tu-braunschweig.de/receive/dbbs_mods_00024412, 2008.

115   – *7. The suggested test by BepiColombo relies on the stability of the IMF during the flyby (see also 2 and 3 above). Please comment.*

– Reply:

First of all, there is a correction in the Venus flyby plan for BepiColombo in the manuscript. Venus-Flyby 1 is planned on 12 October 2020 to an altitude down to 11,317 km, and
120         flyby 2 on 11 August 2021 down to 1,000 km. BepiColombo's flyby at Venus is too far to measure the near-surface magnetic field. Yet, the flyby data will help us to determine or constrain the stability of IMF and the condition for the magnetic field penetration through the ionosphere. The test for the magnetic field penetration would ideally be performed during a stable IMF period, for another Venus mission in future.

125   – We will add the above statement in the conclusion section (p. 5, L8–10).

– 8. Minors:

– *p. 1, L9: about ⇒ around?*

– *L12: Orbiter*

– *L14: studied ⇒ observed?*

130   – *L16-17: The field magnitude becomes diminished ⇒ Further down, the field magnitude decreases*

– *L21: as a proof ⇒ as very accurate?*

– *p. 2, L3: becomes reset when the external field (in the induced magnetic field) reverses its orientation ⇒ please rephrase (see also 2).*

135 – *L8: report our study ⇒ find?*

– *L10: delete by permeability. For the substance of the message, see also 2, 3.*

– *L17: L is,..., \mu_0 ... is, and \sigma is...*

– *L20: strictly ⇒ exactly*

– *L25: I think (or diffusion speed) should be deleted.*

140 – *p. 3, L3: zˆ2 ⇒ \Delta zˆ2 (i.e. z_maxˆ2 – z_minˆ2); ⟨\sigma⟩ ⇒ \sigma (no brackets); diffusivity ⇒ conductivity.*

– *L4-5: Please rephrase Since the Pedersen conductivity transmits the magnetic field by diffusion (see also 4).*

– *L20: from Venus Express*

145 – *L31: referred ⇒ inferred*

– Reply:

Thank you for the suggestions. We will add changes in the revision.

– **Editor's note** (4 Oct. 2018)

*The only recommendation from my side is to investigate whether there are more recent global*
150 *simulations of Venus-solar wind interaction, that may include another estimation for the diffusion time of the magnetic field, in addition to what Martinecz et al. (2009) provides. From a short bibliographic search, I came up e.g. with the study by Dimmock et al. (Dimmock, A. P., Alho, M., Kallio, E., Pope, S. A., Zhang, T. L., Kilpua, E., et al. (2018). The re-sponse of the Venusian plasma environment to the passage of an ICME: Hybrid simulation*
155 *results and Venus Express observations. Journal of Geophysical Research: Space Physics, 123, 35803601. https://doi.org/10.1029/2017JA024852) which may be of relevance, although I have not explicitly checked whether that paper contains any relevant information about magnetic field diffusion.*

– Reply:

160 To the authors' knowledge, the magnetic diffusion problem at Venus has not yet been studied qualitatively or quantitatively. For example, the global hybrid simulation by Dimmock et al. (2018) indicates the possibility of ionosphere magnetization during the ICME (interplanetary coronal mass ejection) event, but the diffusion time estimate is not given and the simulation deals with a time-dependent phenomenon of ICME.

165 – We will add Dimmock et al. (2018) to the reference list.

**Changes in the revised manuscript**

**Ref.01.01**
page 2, line 31 to page 3, line 2
"It is worth mentioning that ... ionosphere."

**Ref.01.02**
page 2, line 15--17
"Here, we mean by the... ionosphere"

**Ref.01.03**
page 2, line 22--26
"We take ...solar minimum."

**Ref.01.04**
page 3, line 9--18
"There are...into heat."

**Ref.01.05**
page 1, line 3
"between 12 hours and 54 hours"

page 5, Eq. (4)
Only Equation (4) is shown, and Equation (5) in the original manuscript
has been deleted.

page 5, line 8--17
"The ion term ....at about 10 S m^-1."

page 5
Table 1 has been updated.

page 6
Figure 1 has been updated.

page 6, line 2--7
"Diffusion time... days"

**Ref.01.06**
page 1, line 19 to page 2, line 2
"Hybrid code simulations....Bößwetter 2009)."

page 8, line 5--9,  line 22--24
Reference to Bößwetter et al. (2004), Bößwetter (2009), and Martinecz (2008).

page 2, line 4--8

"Numerical diffusion... problem."

**Ref.01.07**
page 7, line 3--10
"Third, the upcoming missions...future."

page 8, line 2--4, line 17--19, line 29--30
Reference to Benkhoff et al. (2010), Fox et al. (2016), and Müller et al. (2013).

**Ref.01.08**
page 1, line 9, "around"
page 1, line 12, "Orbiter"
page 1, line 14, "observed"
page 1, line 16--17, "Further down, the field magnitude decreases"
page 2, line 4, "as very accurate"
page 2, line 15--17, (We keep the term "reset" and add an explanation in the text.)
page 2, line 27, "find"
page 2, line 29, "by permeability" has been deleted. "Venus interior for a long time period"
page 3, line 7, "is" (three times)
page 3, line 21, "exactly"
page 3, line 26, "(or diffusion speed)" has been deleted. "differential diffusion time"
page 4, line 3, $L^2 \mu_0 \sigma$,
page 4, line 3, "conductivity"
page 4, line 4--5, "Since we work on the Pedersen conductivity
              for the magnetic diffusion problem"
page 4, line 20, "from Venus Express"
page 4, line 31, "inferred"

**Others**
page 7, line 13--14
"YN thanks... plan."

**Editor's note**
page 2, line 8--9
"A more recent... (Dimmock et al., 2018)."

page 8, line 10--12
Reference to Dimmock et al. (2018)

---

## Referee Report (RR1)

In general, I am happy with the adjustments made to the paper. The authors may still want to consider a number of issues, listed below, that require further clarification.

1. When I first read the manuscript I was quite surprised by the difference between the simulation and analytic estimates of the diffusion time, of about two orders of magnitude. From the revised manuscript, I understand that the major reason behind is the neglect of the electron-neutral collisions in the hybrid simulations (and related contribution to the conductance), with additional impact of numerical diffusion (p.2, L4–9). According to the Response to my comments (L79), including the ion term in the conductance estimate results in a diffusion time longer by 10-20%. Conversely, the electron contribution to the conductance (and diffusion time) is some 80-90%, i.e. by neglecting this contribution the diffusion time is 5 to 10 times shorter than its actual value. Therefore, considering the electron contribution would increase the simulation proxy of 1000 s (p.2, L2) to some 5000–10000 s, i.e., 1.5–3 h. This reduces the difference between the simulation and analytic result by one order of magnitude. While my expertise on simulations is limited and I cannot fully judge the importance of numerical diffusion, I find this smaller difference more reasonable. A more detailed discussion (e.g., in the Intro and/or in Section 2.2), perhaps including the quantitative aspect above, may help the reader to get the message better.

2. I would appreciate as well a more comprehensive discussion of the conductivity estimate (which controls, essentially, the diffusion time), including a comparison with the Earth case. This would cast the matter into a broader perspective and help pointing out the (different) parameter regime at Venus. In particular, the actual value of the conductivity is much higher than at the Earth, by about four orders of magnitude, which has to do, I guess, with the weaker magnetic field and higher neutral density. At the same time, the Pedersen conductance appears to be dominated by electrons, unlike at the Earth, where is dominated by ions, with important consequences, e.g. on the simulation results (as commented above).

3. The newly introduced para at p. 3, L9–18, is somewhat confuse:
 3.1. First, I am not sure if the geometric aspect, emphasized there, is the key one for the diffusive process. I think the energetic aspect is at least as important, since diffusion takes time because energy is dissipated along the way (magnetic energy is converted to heat). In the extreme case where conductivity is (quasi)zero, magnetic field penetration is (quasi)instantaneous. I think the energetic aspect, not the geometric one, is the main reason behind using the Pedersen conductance to estimate the diffusion time. Energy dissipation is achieved by Pedersen current, while Hall current has no energetic effect.
 3.2. Second, in order to clarify better the geometric aspect, the average orientation of the magnetic field in the (current carrying layer of the) ionosphere should be indicated. From the text, I infer that magnetic field is essentially included in the ionospheric plane (like at the Earth equator). Correct?

4. Height integration (point 5 of Section 2.2) and conductivity panel of Fig. 1: My guess is that most of the conductance (height integrated conductivity) is confined around the maximum at 140 km, perhaps within 20 to 40 km. Can you indicate what is the (percentagewise) contribution of this (current carrying) layer, and what of the rest? (including the altitude regions, low and high, subject to extrapolation)

5. Minors:
- p.2, L1: simulation *time* as
-      L3: numerical diffusion superimposes the physical diffusion => numerical resistivity significantly exceeds physical resistivity (?)
-      L9: the ICME => an ICME
-      L16: linear to the => linear in the
-      L17: the magnetic energy stored in => magnetic energy supplied to (?)
-      L23: for the reason of => to infer
-      L26: Delete 'even' and 'during the solar minimum'.
-      L30: Correct 'down to 90 km' (1000 km, according to p7, L7); by the Bepi => during the Bepi
-      L33: problem into => problem to
- p.3, L9: There are three different kinds of conductivity in the plasma => In general, conductivity in a magnetized plasma is a tensor, whose components are
- p.4, L16–18 and caption of Fig. 1: Please refer also to ion-neutral collisions, as included in the revised manuscript.
- p.7, L5: in a variety => for a variety; exampl,e => example;
-      L5–6: BepiColombo plans two Venus flyby maneuvers => two Venus flyby manoeuvres are planned for BepiColombo
-      L6: Delete 'to an altitude'.
-      L8–9: How long does it take the flyby?
- Ack: Venur

---

## Author Response (AR2)

Manuscript prepared for Ann. Geophys.
with version 4.2 of the LATEX class copernicus.cls.
Date: 1 November 2018

**Reply to referee comments – angeo-2018-69 (revision 2)**

Y. Narita[1,2] and U. Motschmann[3,4]

[1]Space Research Institute, Austrian Academy of Sciences, Schmiedlstr. 6, A-8042 Graz, Austria
[2]Institute of Physics, University of Graz, Universitätsplatz 5, A-8010 Graz, Austria
[3]Institut für Theoretische Physik, Technische Universität Braunschweig, Mendelssohnstr. 3, D-38106 Braunschweig, Germany
[4]Deutsches Zentrum für Luft- und Raumfahrt, Institut für Planetenforschung, Rutherfordstr. 2, D-12489 Berlin, Germany

*Correspondence to:* Y. Narita
(yasuhito.narita@oeaw.ac.at)

- *In general, I am happy with the adjustments made to the paper. The authors may still want to consider a number of issues, listed below, that require further clarification.*

  *1. When I first read the manuscript I was quite surprised by the difference between the simulation and analytic estimates of the diffusion time, of about two orders of magnitude. From the revised manuscript, I understand that the major reason behind is the neglect of the electron-neutral collisions in the hybrid simulations (and related contribution to the conductance), with additional impact of numerical diffusion (p.2, L4-9). According to the Response to my comments (L79), including the ion term in the conductance estimate results in a diffusion time longer by 10–20 %. Conversely, the electron contribution to the conductance (and diffusion time) is some 80–90 %, i.e. by neglecting this contribution the diffusion time is 5 to 10 times shorter than its actual value. Therefore, considering the electron contribution would increase the simulation proxy of 1000 s (p.2, L2) to some 500010000 s, i.e., 1.5-3 h. This reduces the difference between the simulation and analytic result by one order of magnitude. While my expertise on simulations is limited and I cannot fully judge the importance of numerical diffusion, I find this smaller difference more reasonable. A more detailed discussion (e.g., in the Intro and/or in Section 2.2), perhaps including the quantitative aspect above, may help the reader to get the message better.*

  - Reply: Good idea. We added an explanation as to why our estimate of the diffusion time is so different from the hybrid simulation by two orders of magnitude. Absence of electron-neutral collisions is certainly one of the reasonable explanations. Following the suggestion by the referee, we performed an order-of-magnitude estimate for the artificial or numerical diffusion. We found out that the numerical diffusion originates in the spatial smoothing procedure to stabilize numerical instabilities (oscillations and divergence). The smoothing procedure in the simulation can explain the difference in the conductivity estimate by about two orders of magnitude. So, both the absence of electron-neutral collisions and the spatial smoothing in the simulation can reasonably explain the difference in the conductivity estimate.

– We added the following text in section 2.3.

Page 7 line 2–6.

"For reference, we repeat the calcuation of the diffution time using only the electron term in the conductivity. The diffusion time from the electron term is in the range from about 40,000 s (11 hours) to about 146,000 s (40 hours). The ion contribution makes a difference in the diffusion time by about 10–20 %. Note that the peak Pedersen conductivity is about $14$ S m$^{-1}$ from the electron term and about $0.035$ S m$^{-1}$ (Tab. 2 right column), i.e., the ion contribution is only about $0.2$ % at the peak of the Pedersen conductivity height-profile."

– We also added a new section (Sec. 3 "Discussion") and discuss the difference from the simulation results in new section 3.1. Accordingly, the conclusion section was moved to new section 4 "Concluding remark".

Page 7, line 8 to page 8, line 17.

"It is interesting to observe the difference between the simulation and analytic estimates of the diffusion time by about two orders of magnitude. The major reason behind this difference most likely lies in the neglect of the electron-neutral collisions and the numerical diffusion in the hybrid simulations.

- Electron contribution. The ion term in the conductance estimate contributes to a longer diffusion time by 10–20 %. In other words, the electron contribution to the conductance (and diffusion time) is somewhat 80–90 %, i.e., by neglecting this contribution the diffusion time is 5 to 10 times shorter than its actual value. Therefore, considering the electron contribution would increase the simulation proxy of 1000 s to 5000–10000 s, i.e., 1.5–3 hours. This reduces the difference between the simulation and analytic result by one order of magnitude.

- Numerical diffusion. Problem of numerical oscillation, rounding or cutoff error, and numerical instabilities in the computation are suppressed either by adding an articicial diffusion or imposing a spatial smoothing (Winske and Omidi, 1993; Bagdonat, 2002). Here we estimate the diffusion coefficient or the conductivity for the

smoothing procedure. First, we rewrite the magnetic diffusion equation

$$\partial_t \boldsymbol{B} = \eta \nabla^2 \boldsymbol{B} \tag{5}$$

into the time advancing formula as

$$\boldsymbol{B}(t + \Delta t) = \eta \nabla^2 \boldsymbol{B}(t)\,\Delta t \tag{6}$$

$$\simeq \frac{\eta}{\ell^2} \boldsymbol{B}(t)\,\Delta t, \tag{7}$$

where $\eta$ denotes the diffusion coefficient of the magnetic field, $\boldsymbol{B}$ is the magnetic field, $\Delta t$ is the time step in the simulation, and $\ell$ is the length scale of the gradient. Second, in the smoothig method, the magnetic field is smoothed by the following procedure (Winske and Omidi, 1993; Bagdonat, 2002; Müller et al., 2011),

$$\boldsymbol{B}(t) \to \boldsymbol{B}(t) - \alpha_{\mathrm{sm}}\left(\boldsymbol{B}(t) - \langle \boldsymbol{B} \rangle\right), \tag{8}$$

where $\alpha_{\mathrm{sm}}$ is a free parameter called the the smoothing factor (its value of $\alpha_{\mathrm{s}}$ is typically 0.01 to 0.1) and $\langle \boldsymbol{B} \rangle$ is the locally-smoothed magnetic field. Now, by comparing the right-hand side of Eq. (7) with the smoothing term $\alpha_{\mathrm{sm}}\boldsymbol{B}$ in Eq. (8), we obtain a relation between the smoothing factor $\alpha_{\mathrm{sm}}$ and the diffusion coefficient $\eta$ as follows,

$$\eta = \frac{\alpha_{\mathrm{sm}}\ell^2}{\Delta t}. \tag{9}$$

Using the relation to the conductivity $\eta = (\mu_0 \sigma)^{-1}$, Eq. (9) is expressed as

$$\sigma = \frac{\Delta t}{\mu_0 \alpha_{\mathrm{sm}} \ell^2}. \tag{10}$$

We evaluate Eq. (10) using the following values: the smoothing parameter $\alpha_{\mathrm{sm}} = 0.01$ (i.e., 1 % spatial smoothing) from Müller et al. (2011), $\ell = 100$ km (grid size in the simulation) from Bößwetter et al. (2004) and Martinecz et al. (2009), $\Delta t = 1$ s (typical ion gyroperiod in the solar wind), and $\mu_0 = 4\pi \times 10^{-7}$ H m$^{-1}$ (permeability of free space), and obtain the numerical conductivity (or the smoothing conductivity) $\sigma_{\mathrm{sm}} = 7.96 \times 10^{-3}$ S m$^{-1}$. Hence, the numerical conductivity equivalent to the spatial smoothing procedure for the purpose of numerical damping in the hybrid simulation is of the order of $10^{-2}$ S m$^{-1}$. The physical conducticity from our study is 1–10 S m$^{-1}$ at the peak.

The magnetic diffusion time is $L^2 \mu_0 \sigma$ and the diffusion length scale $L$ is the same as the grid scale $l$. Hence, the diffusion time is proportional to the conductivity in our problem, and the difference in the diffusion time by two orders of magnitude between the hybrid simulations (about 1000 s) and our semi-analytic estimate (about 100,000 s) can reasonably be explained by the spatial smoothing procedure in the simulation."

– 2. *I would appreciate as well a more comprehensive discussion of the conductivity estimate (which controls, essentially, the diffusion time), including a comparison with the Earth case. This would cast the matter into a broader perspective and help pointing out the (different) parameter regime at Venus. In particular, the actual value of the conductivity is much higher than at the Earth, by about four orders of magnitude, which has to do, I guess, with the weaker magnetic field and higher neutral density. At the same time, the Pedersen conductance appears to be dominated by electrons, unlike at the Earth, where is dominated by ions, with important consequences, e.g. on the simulation results (as commented above).*

– Reply: The most essential difference is the lack of strong magnetic field at the Venus. This leads to a lower gyro-frequency at higher altitudes and the electron-neutral collision frequency becomes dominant at altitudes of 150 km. The electron density is nearly the same ($10^5$ cm$^{-3}$) between the two planets at the relevant altitudes (150 km at the Venus, 130 km at the Earth). So, the essential difference in the conductivity estimate between the Venus and the Earth comes from the ion-to-electron mass ratio (about 20,000) and this factor is slightly compensated by the ratio of the electron-neutral collision frequency at the Earth to the ion-neutral collision frequency at the Venus (a factor of about 0.2). In summary, the ratio of the peak conductivity at the Venus to that at the Earth is in the range between 6,000 and 10,000.

– We added a new subsection 3.2 "Comparison with the Earth ionosphere".

Page 8, line 18 to page 9, line 17.

"It is also interesting to observe that the Pedersen conductivity at the Venus is much higher than at the Earth by about four orders of magnitude. This difference can be explained as follows. We write formulas separately for the electron Pedersen conductivity $\sigma_{\mathrm{p,e}}$ S m$^{-1}$ as

$$\sigma_{\mathrm{p,e}}[\mathrm{Sm}^{-1}] = \frac{n_{\mathrm{e}} e^2}{m_{\mathrm{e}}} \frac{\nu_{\mathrm{en}}}{\nu_{\mathrm{en}}^2 + f_{\mathrm{ge}}^2} \tag{11}$$

$$\simeq 2.8 \times 10^{-2} \times n_{\mathrm{e}}[\mathrm{cm}^{-3}] \times \frac{\nu_{\mathrm{en}}[\mathrm{Hz}]}{(\nu_{\mathrm{en}}[\mathrm{Hz}]) + (f_{\mathrm{ge}}[\mathrm{Hz}])^2} \tag{12}$$

and the ion Pedersen conductivity $\sigma_{\mathrm{p,i}}$ S m$^{-1}$ as

$$\sigma_{\mathrm{p,i}}[\mathrm{Sm}^{-1}] = \frac{n_{\mathrm{e}} e^2}{m_{\mathrm{i}}} \frac{\nu_{\mathrm{in}}}{\nu_{\mathrm{in}}^2 + f_{\mathrm{gi}}^2} \tag{13}$$

$$\simeq 1.3 \times 10^{-6} \times n_{\mathrm{e}}[\mathrm{cm}^{-3}] \times \frac{\nu_{\mathrm{in}}[\mathrm{Hz}]}{(\nu_{\mathrm{in}}[\mathrm{Hz}])^2 + (f_{\mathrm{gi}}[\mathrm{Hz}])^2}, \tag{14}$$

where a mass ratio of $m_{\mathrm{i}}/m_{\mathrm{p}} = 11.6$ or $m_{\mathrm{i}}/m_{\mathrm{e}} = 2.1 \times 10^4$ is used ($m_{\mathrm{i}}$ is the ion mass, $m_{\mathrm{p}}$ is the proton mass, and $m_{\mathrm{e}}$ is the electron mass).

The Pedersen conductivity at the Venus $\sigma_{\mathrm{p}}^{(\mathrm{V})}$ is primarily contributed by the electron-neutral colissions. By ignoring the gyro-frequency, the Venus Pedersen conductivity is

approximately as follows.

$$\sigma_{\mathrm{p}}^{(\mathrm{V})} \sim \frac{n_{\mathrm{e}} e^2}{m_{\mathrm{e}} \nu_{\mathrm{en}}} \tag{15}$$

The Pedersen conductivity at the Earth $\sigma_{\mathrm{p}}^{(\mathrm{E})}$ is, in contrast to the Venus case, contributed by the ion-neutral collisions.

$$\sigma_{\mathrm{p}}^{(\mathrm{E})} \sim \frac{n_{\mathrm{e}} e^2}{m_{\mathrm{i}} \nu_{\mathrm{in}}} \tag{16}$$

The reason for this is that the gyro-frequency exceeds the electron-neutral collision frequency at an altitude of 60–70 km and above due to a stronger magnetic field (than that of the Venus). Now we compare Eq. (15) with Eq. (16).

Using the facts that (1) the electron density is almost the same ($n_{\mathrm{e}} \sim 10^5\,\mathrm{cm}^{-3}$) between the Venus (peak altitude $z = 150\,\mathrm{km}$) and the Earth (peak altitude $z = 130\,\mathrm{km}$), (2) the typical mass ratio from the ions to the electrons is about $20{,}000$ (about 11 proton mass), and (3) the collision frequency is rougfhly of the same order beween the Venus and the Earth, $\nu_{\mathrm{en}}^{(\mathrm{V})} = 250\,\mathrm{Hz}$ and $\nu_{\mathrm{in}}^{\mathrm{E}} = 50\,\mathrm{Hz}$ at the Earth, respectively, we obtain the ratio of the peak Pedersen conductivity from the Venus to the Earth as follows:

$$\frac{\sigma_{\mathrm{p,e}}^{(\mathrm{V})}}{\sigma_{\mathrm{p,i}}^{(\mathrm{E})}} \sim \frac{m_{\mathrm{i}}}{m_{\mathrm{e}}} \frac{\nu_{\mathrm{in}}^{(\mathrm{E})}}{\nu_{\mathrm{en}}^{(\mathrm{V})}} \sim 10^4. \tag{17}$$

The difference in the peak Pedersen conductivity by nearly four orders of magnitude can essentially represent the difference in the Pedesen current carrier in the different magnetic field environments: the Pedersen current is carried by the electrons at the Venus and by the ions at the Earth. More detailed calculations of the peak Pedersen conductivity at the Earth and the Venus are shown in Tab. 2."

– Table 2 was added, too.

**Table 2.** Comparison of Pedersen conductivity estimate at the peak altitudes of the conductivity, $z = 130$ km at the Earth and $z = 150$ km at the Venus. Electron density value at the Earth is taken from Kelly (1989) at $z = 130$ km. Electron density value at the Venus is from Kilore and Luhmann (1991) during the solar maximum at an altitude of $z = 150$ km (at the peak of Pedersen conductivity). Electron gyro-frequency is calculated from the nominal magnetic field magnitude. Collision frequency values at the Earth are from Kertz (1989) and that at the Venus are from Dubinin et al. (2014). A mean mass ratio of 11.6 is used between the ions and the protons.

|  | Earth | Venus |
|---|---|---|
| magnetic field $B$ | $3.0 \times 10^5$ nT | $1.0 \times 10$ nT |
| electron density $n_e$ | $1.2 \times 10^5$ cm$^{-3}$ | $2.9 \times 10^5$ cm$^{-3}$ |
| electron gyro-frequency $f_{ge}$ | $8.4 \times 10^4$ Hz | $2.8 \times 10^2$ Hz |
| electron-neutral collision frequency $\nu_{en}$ | $1.3 \times 10^3$ Hz | $2.5 \times 10^2$ Hz |
| electron Pedersen conductivity $\sigma_{p,e}$ | $6.2 \times 10^{-4}$ S m$^{-1}$ | $1.4 \times 10^1$ S m$^{-1}$ |
| ion gyro-frequency $f_{gi}$ | $3.9 \times 10^1$ Hz | $1.3 \times 10^{-2}$ Hz |
| ion-neutral collision frequency $\nu_{in}$ | $5.0 \times 10^1$ Hz | $1.1 \times 10$ Hz |
| ion Pedersen conductivity $\sigma_{p,i}$ | $2.3 \times 10^{-3}$ S m$^{-1}$ | $3.5 \times 10^{-2}$ S m$^{-1}$ |

115     – *3. The newly introduced para at p. 3, L9–18, is somewhat confuse:*

*3.1. First, I am not sure if the geometric aspect, emphasized there, is the key one for the diffusive process. I think the energetic aspect is at least as important, since diffusion takes time because energy is dissipated along the way (magnetic energy is converted to heat). In the extreme case where conductivity is (quasi)zero, magnetic field penetration is (quasi)instantaneous. I*

120     *think the energetic aspect, not the geometric one, is the main reason behind using the Pedersen conductance to estimate the diffusion time. Energy dissipation is achieved by Pedersen current, while Hall current has no energetic effect.*

*3.2. Second, in order to clarify better the geometric aspect, the average orientation of the magnetic field in the (current carrying layer of the) ionosphere should be indicated. From*

125     *the text, I infer that magnetic field is essentially included in the ionospheric plane (like at the Earth equator). Correct?*

     – Reply:

3.1 Agreed. We added the energy dissipation aspect of the conductivity before the geometrial aspect.

130     3.2 Yes.

     – Changes in the manuscript:

Page 3, line 10–13.

"Above all, the Pedersen conductivity is relevant to the problem of diffusion time esti-

135     mate. The reason for this is that magnetic diffusion takes time because energy is dissipated along the way (magnetic energy is converted to heat). It is the Pedersen current by which the energy dissipation is achieved. The Hall current, in contrast, has no energetic effect. From a geometrical point of view,"

140     Page 3, line 14–15

"or in the current-carrying layer of the ionosphere"

    – *4. Height integration (point 5 of Section 2.2) and conductivity panel of Fig. 1: My guess is that most of the conductance (height integrated conductivity) is confined around the maximum at 140 km, perhaps within 20 to 40 km. Can you indicate what is the (percentagewise) contri-*

145     *bution of this (current carrying) layer, and what of the rest? (including the altitude regions, low and high, subject to extrapolation)*

     – Reply:

Yes, the most of the conductivity is confied to a layer between an altitude of 140 km and 170 km. Maximum of the conductivity is around an altitude of 150 km. We calculated

150     the contribution of this layer to the condutivity, and obtain a percentage of 78–85 %

during the solar maximum and 88–91 % during the solar minimum.

– Changes in the manuscript:

Page 5, line 27–30.

"Most of the conductance (height-integrated conductivity) is confined around the maximum at 150 km. The contribution of this current-carrying layer from 140 km to 170 km in altitude is around 78–85 % during the solar maximum (the variation comes from the choice of the magnetic field mode and the ion profile) and 88–91 % during the solar minimum."

– *5. Minors:*

*- p.2, L1: simulation \*time\* as*

*- L3: numerical diffusion superimposes the physical diffusion ⇒ numerical resistivity significantly exceeds physical resistivity (?)*

*- L9: the ICME ⇒ an ICME*

*- L16: linear to the ⇒ linear in the*

*- L17: the magnetic energy stored in ⇒ magnetic energy supplied to (?)*

*- L23: for the reason of ⇒ to infer*

*- L26: Delete even and during the solar minimum.*

*- L30: Correct down to 90 km (1000 km, according to p7, L7); by the Bepi ⇒ during the Bepi*

*- L33: problem into ⇒ problem to*

*- p.3, L9: There are three different kinds of conductivity in the plasma ⇒ In general, conductivity in a magnetized plasma is a tensor, whose components are*

*- p.4, L16–18 and caption of Fig. 1: Please refer also to ion-neutral collisions, as included in the revised manuscript.*

185     *- p.7, L5: in a variety ⇒ for a variety; exampl,e ⇒ example;*

*- L5–6: BepiColombo plans two Venus flyby maneuvers ⇒ two Venus flyby manoeuvres are planned for BepiColombo*

190     *- L6: Delete to an altitude.*

*- L8–9: How long does it take the flyby?*

*- Ack: Venur*
195

     **–** Reply:
      Done.

     **–** Changes in the manuscript:
      Page 1, line 23, "simulation time"
200      Page 2, line 2, "numerical resistivity..."
      Page 2, line 7, "an ICME"
      Page 2, line 15, "linear in the"
      Page 2, line 16, "magnetic energy supplied to the ionosphere"
      Page 2, line 22, "to infer"
205      Page 2, line 24–25, "which shortens..."
      Page 2, line 29–30, "down to 1000 km..."
      Page 2, line 33, "problem to"
      Page 3, line 9, "In general..."
      Page 6, figure 1, line 2, "and that..."
210      Page 9, line 25, "example"
      Page 9, line 25, "two Venus flyby manoeuvres are..."
      Page 9, line 26, "down to"
      Page 9, line 29, "for several hours to a day"
      Page 10, Acknowledgments, line 3, "Venus"

215      **–** The following references were added.

       • Bagdonat, T. B.: Hybrid simulations of weak comets, PhD thesis, Tech. Univ. Braunschweig, Braunschweig, 2002.

       • Kelly, M. C.: The Earths Ionosphere, Academic Press, Inc., San Diego, 1989.

       • Kertz, W.: Einführung in die Geophysik II, BI Hochschultaschenbücher, 1989.

220       • Müller, J., Simon, S., Motschmann, U., Schüle, J., Glassmeier, K.-H., and Pringle, G. J.: A.I.K.E.F.: Adaptive hybrid model for space plasma simulations, Comp. Phys. Comm., 182, 946–966, https://10.1016/j.cpc.2010.12.033, 2011.

• Winske, D., and Omidi, N.: Hybrid codes: Methods and applications, Terra Publication, Tokyo, 1993.